# Optimal anti-amyloid-beta therapy for Alzheimer's disease via a personalized mathematical model

**Wenrui Hao**[1]*, **Suzanne Lenhart**[2], **Jeffrey R. Petrella**[3]

**1** Department of Mathematics, Penn State University, University Park, Pennsylvania, United States of America, **2** Department of Mathematics, University of Tennessee, Knoxville, Tennessee, United States of America, **3** Department of Radiology, Duke University Health System, Durham, North Carolina, United States of America

* wxh64@psu.edu

**Data Availability Statement:** There are no primary data in the paper; all data is available at https://adni.loni.usc.edu/ and our code is published through https://github.com/whao2008/AD_optimal_control.

## Abstract

With the recent approval by the FDA of the first disease-modifying drug for Alzheimer's Disease (AD), personalized medicine will be increasingly important for appropriate management and counseling of patients with AD and those at risk. The growing availability of clinical biomarker data and data-driven computational modeling techniques provide an opportunity for new approaches to individualized AD therapeutic planning. In this paper, we develop a new mathematical model, based on AD cognitive, cerebrospinal fluid (CSF) and MRI biomarkers, to provide a personalized optimal treatment plan for individuals. This model is parameterized by biomarker data from the AD Neuroimaging Initiative (ADNI) cohort, a large multi-institutional database monitoring the natural history of subjects with AD and mild cognitive impairment (MCI). Optimal control theory is used to incorporate time-varying treatment controls and side-effects into the model, based on recent clinical trial data, to provide a personalized treatment regimen with anti-amyloid-beta therapy. In-silico treatment studies were conducted on the approved treatment, aducanumab, as well as on another promising anti-amyloid-beta therapy under evaluation, donanemab. Clinical trial simulations were conducted over both short-term (78 weeks) and long-term (10 years) periods with low-dose (6 mg/kg) and high-dose (10 mg/kg) regimens for aducanumab, and a single-dose regimen (1400 mg) for donanemab. Results confirm those of actual clinical trials showing a large and sustained effect of both aducanumab and donanemab on amyloid beta clearance. The effect on slowing cognitive decline was modest for both treatments, but greater for donanemab. This optimal treatment computational modeling framework can be applied to other single and combination treatments for both prediction and optimization, as well as incorporate new clinical trial data as it becomes available.

## Author summary

Although personalized therapy will likely play a major role in the appropriate management and counseling of patients with AD in the future, there are currently no clinically

**Funding:** WH was supported in part by National Science Foundation (NSF) DMS-2052685 (https://nsf.gov/). JRP was supported in part by NSF DMS-2052676 (https://nsf.gov/). The funders had no role in study design, data collection and analysis, decision to publish, or preparation of the manuscript.

**Competing interests:** The authors have declared that no competing interests exist.

utilized markers that can easily distinguish among the different clinical trajectories of individual patients, nor provide personalized treatment plans. The mathematical model developed in this paper, based on current theories of AD pathophysiology, enables prediction of disease trajectory under a natural history scenario in individual patients with a clinical diagnosis of AD or late MCI (L-MCI) using current clinically validated biomarkers. This analytical approach also provides an in-silico method to simulate and optimize treatment at an individual level, thereby accelerating the development of personalized treatments. By accessing longitudinal biomarker data from the ADNI database, we validate our computational modeling approach to identify patient-specific disease trajectories and optimize individual treatments for two anti-amyloid-beta therapies, aducanumab and donanemab, in proof-of-principle clinical trial simulations. Simulation results show that, with the optimization, the effect on slowing cognitive decline is greater for doneneumab than aducanumab for a 10-year treatment regimen, although the effect on amyloid beta clearance is similar for both drugs.

## Introduction

Alzheimer's disease (AD) affects more than 5 million people in the U.S. and is recognized as one of the leading global health priorities of the 21st century [1]. On June 7, 2021, the U.S. Food and Drug Administration (FDA) granted accelerated approval for the first-ever disease-modifying therapy for AD, aducanumab, a monoclonal antibody directed against amyloid-beta protein. This therapy has been shown to effectively remove amyloid plaques from the brain. Still, however, questions remain regarding the efficacy of removing amyloid plaques for preventing or delaying cognitive decline [2]. This uncertainty, combined with the 99% failure rate of trials of other classes of AD treatments, is rooted in an incomplete understanding of the complex mechanisms resulting in AD, and how disease trajectory and response to treatment may vary individual-to-individual. It is likely, therefore, that personalized treatment will need to play a central role in the future management and counseling of patients with AD [3, 4]. Tailored approaches to treatment will be facilitated by the growing availability of electronic data in AD subjects and those at risk. Two components are necessary to realize this idea: first, an abundance of longitudinal data to cover many physiological aspects of individuals when they are healthy and possibly into disease [5]; second, computational methods and models capable of analyzing and integrating this data on a large scale [6].

Although computational modeling is still an emerging field in the study of AD, several mathematical models have been developed based on systems biology approaches to AD molecular and cellular patho-physiologic mechanisms. Our group, for example, built a model based on AD signaling pathways using a system of partial differential equations (PDEs) [7]. This model has been used to simulate and validate at a cellular level the mechanisms underlying the failure of several drugs in recent clinical trials. Because the variables in this and similar mechanistic models cannot be measured directly in living subjects, simulated treatment studies can only be performed at the population, rather than individual patient, level. Treatment dosage and regimen, therefore, might not be optimal for each individual. Over the past two decades, several clinical biomarkers of AD patho-physiologic progression have been developed to track disease progression in patient-oriented research. Broadening our previous mathematical modeling approach, based on molecular and cellular mechanisms, to these key AD clinical biomarkers, we developed a sparse cascade model to include pathologic hallmark biomarkers

**Fig 1. Flowchart of personalized optimal therapeutic study: Starting with the ODE cascade model, we calibrate individual parameters using longitudinal biomarker data for each subject in the ADNI dataset.** Optimal control theory is then applied to the personalized models with treatment as a control function to simulate both short-term (78 weeks) and long-term (10 years) optimized digital clinical trials initiated at chronological ages 60 and 70. Trials are conducted for the anti-amyloid-beta agents, aducanumab at two different doses, and for donanemab at a single dose.

(amyloid beta and tau), neuronal loss biomarkers, and cognitive impairment using a system of ordinary differential equations (ODEs) [8].

In this paper, we develop a novel personalized treatment optimization framework based on a mathematical modeling approach. Our contribution is the following:

- we develop a sparse empirical cascade model of AD progression to include only clinical biomarkers of beta-amyloid and taupathology, neuronal degeneration, and cognitive impairment;

- we parametrize the model on a multicenter dataset with available cerebrospinal fluid (CSF), MRI and cognitive biomarkers to build a personalized model for each individual;

- we perform personalized therapeutic simulation studies for AD and late mild cognitive impairment (LMCI) subjects via application of optimal control theory and corresponding numerical results with our mathematical model.

We apply this modeling framework to conduct in-silico clinical trials of two anti-amyloid-beta treatments using personalized optimal treatment regimens for each individual. This optimal control application allows for time-varying controls [9] to achieve a desired goal to minimize cognitive impairment and the level of amyloid in the brain while minimizing side effects, particularly early in the treatment when they are more likely to occur. Although this approach has been used in various disease treatment models, [10–12] this is a novel application of optimal control theory to treatment of Alzheimer's disease employing personalized regimens. The flowchart of the personalized optimal treatment study is shown in Fig 1.

## Materials and methods

### Mathematical model

In this paper, we develop a cascade model including four AD clinical biomarkers: pathologic hallmark biomarkers (amyloid beta and tau), neuronal loss biomarkers, and cognitive impairment. The pathophysiological network of AD starts with amyloid beta in soluble form and in plaques. This promotes the abnormal phosphorylation of tau protein, leading to neurodegeneration, and finally, via large-scale brain network disruption, to cognitive impairment shown in Fig 2.

**Amyloid beta equation.** The sentinal event in AD is thought to result from an imbalance in $A_\beta$ production and clearance, leading to amyloid plaque accumulation. $A_\beta$ accumulation is

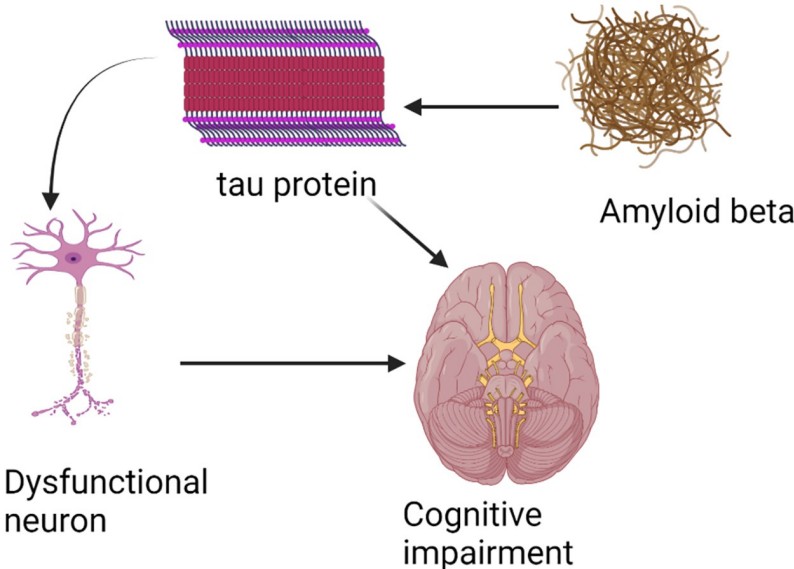

**Fig 2. The biomarker cascade in AD starts with amyloid beta pathology.** This leads to amyloid-related tau pathology, neuronal dysfunction/loss and subsequent cognitive impairment.

modeled by logistic growth [13], namely,

$$\frac{dA_\beta}{dt} = \lambda_{A_\beta} A_\beta \left(1 - \frac{A_\beta}{K_{A_\beta}}\right) \text{ with } A_\beta(T_0) = A_0, \qquad (1)$$

where $A_0$ is the initial condition of amyloid beta at age $T_0$ and may vary for different patients. Here $K_{A_\beta}$ is carrying capacity and $\lambda_{A_\beta}$ is the $A_\beta$ growth rate. The analytical solution of $A_\beta$ is obtained by solving the differential equation, namely,

$$A_\beta(t) = \frac{K_{A_\beta}}{C_1 e^{-\lambda_{A_\beta}(t-T_0)} + 1} \text{ where } C_1 = \frac{K_{A_\beta}}{A_o} - 1.$$

**Tau equations.** Numerous studies of the pathological changes that characterize AD show that amyloid beta accumulation initiates phosphorylation of tau protein [14, 15]. Thus we take the equation of phosphorylated tau, $\tau_p$, as

$$\frac{d\tau_p}{dt} = \lambda_\tau A_\beta \left(1 - \frac{\tau_p}{K_{\tau_p}}\right) \text{ with } \tau_p(T_0) = \tau_{p0}. \qquad (2)$$

Moreover, there may also be nonamyloid-dependent tau accumulation, in which case the cascade mainly depends on comorbidities, e.g., aging and/or suspected non-Alzheimer pathology (SNAP) via nonamyloid-dependent tauopathy, $\tau_o$. We assume that $\tau_o$ linearly grows with respect to age and take

$$\frac{d\tau_o}{dt} = \lambda_{\tau_o} \text{ with } \tau_o(T_0) = \tau_{o0}. \qquad (3)$$

**Neurodegeneration equation.** Tau deposition within cells disrupts microtubules, impairing axonal transport. P-tau impairs mitochondria and translocates into the nucleus [13, 16]. Thus the total tau induces the neurodegeneration, $N$, accordingly, we have the following equation for $N$

$$\frac{dN}{dt} = (\lambda_{N\tau_o}\tau_o + \lambda_{N\tau_p}\tau_p)\left(1 - \frac{N}{K_N}\right) \text{ with } N(T_0) = N_0.$$ (4)

**Cognitive decline equation.** Initiation of cognitive decline, $C$, is directly determined by both neurodegeneration, $N$ and tau pathology [17, 18]. Therefore we have the equation for $C$ below

$$\frac{dC}{dt} = (\lambda_{CN}N + \lambda_{C\tau}\tau_p)\left(1 - \frac{C}{K_C}\right) \text{ with } C(T_0) = C_0.$$ (5)

Thus we summarize the mathematical model (for our state variables) as a system of ODEs below

$$\begin{cases} \dfrac{dA_\beta}{dt} = \lambda_{A_\beta}A_\beta\left(1 - \dfrac{A_\beta}{K_{A_\beta}}\right) \\[2mm] \dfrac{d\tau_p}{dt} = \lambda_\tau A_\beta\left(1 - \dfrac{\tau_p}{K_{\tau_p}}\right) \\[2mm] \dfrac{d\tau_o}{dt} = \lambda_{\tau_o} \\[2mm] \dfrac{dN}{dt} = (\lambda_{N\tau_o}\tau_o + \lambda_{N\tau_p}\tau_p)\left(1 - \dfrac{N}{K_N}\right) \\[2mm] \dfrac{dC}{dt} = (\lambda_{CN}N + \lambda_{C\tau}\tau_p)\left(1 - \dfrac{C}{K_C}\right) \end{cases} \text{ with } = \begin{cases} A_\beta(T_0) = A_0 \\ \tau_p(T_0) = \tau_{p0} \\ \tau_o(T_0) = \tau_{o0} \\ N(T_0) = N_0 \\ C(T_0) = C_0 \end{cases}.$$ (6)

## Parameter estimations via ADNI dataset

The Alzheimer's Disease Neuroimaging Initiative (ADNI), a multicenter, prospective, naturalistic study, began in 2003, comprises four sequential studies—ADNI-1, ADNI-GO, ADNI-2, and ADNI-3—which followed subjects between 5-15 years, using genetic, blood- and CSF-based, imaging, and cognitive biomarkers (adni.loni.usc.edu). In this paper, we use biomarker data from a subset of the ADNI dataset, ADNI-1 which enrolled 819 subjects with LMCI, early AD, and cognitively normal elderly controls. The study included baseline MRI, CSF, and cognitive data plus 10 years of follow-up at various intervals for the different biomarkers. CSF beta-amyloid peptide ($A\beta42$), total tau, phosphorylated tau levels at baseline and follow up every 2 years up to 10 years are available in ADNI-1 in a subset of approximately 300-400 subjects to estimate parameters in the equations of $A_\beta$, $\tau_p$, and $\tau_o$. Volumetrics, such as hippocampal volume, and neuropsychological tests, such as the Alzheimer's Disease Assessment Scale (ADAS13) score, are available at one year and six-month intervals, respectively, and are used to estimate parameters in the equations of $N$ and $C$, respectively.

In order to illustrate the numerical algorithm of parameter estimations, for simplicity, we write the ODE system as

$$\frac{d\boldsymbol{x}}{dt} = \boldsymbol{G}(\boldsymbol{x}, \boldsymbol{p}), \text{ where } \boldsymbol{x} = (A_\beta, \tau_p, \tau_o, N, C)^T \in \mathbb{R}^5$$ (7)

and $\boldsymbol{p}$ denotes all the parameters and initial conditions. We estimate the parameters for each patient via solving the optimization problem below:

$$\min_{\boldsymbol{p}} \sum_{i=1}^{N} \|\boldsymbol{x}(t_i; \boldsymbol{p}) - \tilde{\boldsymbol{x}}(t_i)\|_2^2, \tag{8}$$

where $\tilde{\boldsymbol{x}}(t_i)$ stands for the available longitudinal CSF biomarker, volumetrics, and ADAS13 data at some measuring time points $t_i$ and $\boldsymbol{x}(t_i; \boldsymbol{p})$ is the solution of the ODE model for given parameter $\boldsymbol{p}$ at $t_i$. The optimization (8) is a non-convex optimization on high dimensional parameter space thus the initial guess for optimization algorithms is very sensitive to find a good local minimum. In order to find a good initial guess, we estimate the parameters sequentially, namely, equation-by-equation [19], because the ODE model is a natural cascade model. More specifically, we first estimate the parameters in the equation of $A_\beta$, namely, $\lambda_{A_\beta}, K_{A_\beta}$, and the initial condition, $A_0$, by using CSF amyloid beta42 biomarker data to solve the sub optimization problem below

$$\min_{\lambda_{A_\beta}, K_{A_\beta}, A_o} \sum_i (A_\beta(t_i; \lambda_{A_\beta}, K_{A_\beta}) - \tilde{A}_\beta(t_i))^2 \text{ with } A_\beta(T_0) = A_o \tag{9}$$

Of note, CSF levels of $A_\beta$ peptide go down with increasing disease burden, and therefore are a surrogate for $A_\beta$ accumulation in the brain. Once the parameters of the $A_\beta$ equation are estimated, we perform the similar procedure for $\tau_p, \tau_o, N$, and $C$ equations. In this case, this "equation-by-equation" procedure, following the cascade progression of AD, gives a good initial guess of the original optimization problem (8) compared to random initialization. In fact, (8) achieves 0.01 by using the "equation-by-equation" procedure while the best value is 0.03 among 100 random initializations. More details of the parameter estimation are shown in **Algorithm 1**. The optimization solver is *fmincon* in *Matlab* used for solving each sub optimization problem.

**Algorithm 1** Parameter estimation by solving the optimization problem (8).

**Input** biomarker datapoints $\tilde{\boldsymbol{x}}(t_i)$ at time $t_i$ for one patient.

1: Solve (9) to obtain a local minimizer for parameters $\lambda_{A_\beta}^0, K_{A_\beta}^0$ and the initial condition $A_0^0$;

2: Fix $\lambda_{A_\beta}^0, K_{A_\beta}^0$, and $A_0^0$ and solve

$$\min_{\lambda_\tau, K_{\tau_p}, \tau_{p0}} \sum_i (\tau_p(t_i; \lambda_\tau, K_{\tau_p}) - \tilde{\tau}_p(t_i))^2$$

to obtain the parameters $\lambda_\tau^0, K_{\tau_p}^0$ and the initial condition $\tau_{p0}^0$;

3: Solve the optimization problem

$$\min_{\lambda_{\tau_o}, \tau_{o0}} \sum_i (\tau_o(t_i; \lambda_{\tau_o}) - \tilde{\tau}_o(t_i))^2$$

to obtain the parameter value $\lambda_{\tau_o}^0$ and the initial condition $\tau_{o0}^0$;

4: Solve the following optimization for

$$\min_{\lambda_{N\tau_o}, \lambda_{N\tau_p}, K_N, N_0} \sum_i (N(t_i; \lambda_{N\tau_o}, \lambda_{N\tau_p}, K_N) - \tilde{N}(t_i))^2$$

to obtain the parameter values $\lambda_{N\tau_o}^0, \lambda_{N\tau_p}^0, K_N^0$ and the initial condition $N_0^0$;

5: Solve the following optimization for

$$\min_{\lambda_{CN}, K_C, C_0} \sum_i (C(t_i; \lambda_{CN}, K_C) - \tilde{C}(t_i))^2$$

to obtain the parameter values $\lambda_{CN}^0, K_C^0$ and the initial condition $C_0^0$;

```
6: Solve the optimization problem (8) by using all the parameters
   obtained in the previous steps as an initial guess.
Output p*.
```

## Personalized optimal anti-amyloid-beta treatment study

In ongoing clinical trials, researchers have developed and are testing several major classes of AD interventions, including anti-amyloid-beta, anti-tau, neuroprotective and cognitive enhancing interventions. In this study, we model current anti-amyloid clinical trial agents for AD and provide a personalized optimal anti-amyloid-beta treatment plan via the ODE model. This approach can be also applied to other treatment plans. In this section, we perform the optimal control for both the AD and LMCI groups in ADNI by using the personalized parameters for each subject. More specifically, we represent anti-amyloid-beta therapy, as control function $u(t)$, as follows in the first state equation:

$$\frac{dA_\beta}{dt} = \lambda_{A_\beta} A_\beta \left(1 - \frac{A_\beta}{K_{A_\beta}}\right) - u(t)A_\beta. \tag{10}$$

The optimal anti-amyloid-beta intervention is chosen to minimize both cognitive impairment, $C$ and side-effects over the treatment interval $[T_1, T_2]$ as well as minimize cognitive impairment and the level of amyloid by the end of the treatment, as represented in the following objective function:

$$\min_{u \in U} J(u) := \alpha_1 A_\beta(T_2) + \alpha_2 C(T_2) + \int_{T_1}^{T_2} C(t)dt + \int_{T_1}^{T_2} \varepsilon(A_\beta(t), t)u^2(t)dt, \tag{11}$$

with the control set,

$$U = \{u(t) \in L^\infty([T_1, T_2]) \,|\, 0 \le u(t) \le u_{max}\}.$$

The term, $\varepsilon(A_\beta(t), t)u(t)^2$ represents the side-effects of the anti-amyloid-beta treatment relative to its benefit over time. More specifically, $\varepsilon(A_\beta(t), t)$ depends upon both the level of amyloid beta and the time duration of the treatment. The most serious side-effects of anti-amyloid-beta treatment are brain edema and hemorrhage which are thought to result from the removal of amyloid plaques from the walls of blood vessels [20]. This leads to leakage at the endothelial junctions and breakdown of the blood-brain barrier. The extent of these gaps in the blood vessel walls is likely related to the overall amyloid burden of the patient and the rate of removal of amyloid, the latter a function of the drug dosage [21]. Because the side effects of aducanumab are more likely observed if a high dose is given in patient with a high amyloid burden [22], we also assume the side effects decay with the time of treatment, in keeping with clinical trial data, and represented by

$$\varepsilon(A_\beta(t), t) = \varepsilon_0 A_\beta(t)e^{-\gamma t}.$$

We seek to find an optimal control $u^\star$ such that

$$J(u^\star) = \min_{u \in U} J(u).$$

Note that the controls and the state variables and their derivatives are uniformly bounded in $L^\infty$ and the problem is convex in the control, which can used to obtain the existence of an optimal control, [23, 24] and thus we can apply Pontryagin's Maximum Principle for the necessary conditions below [25].

By denoting $f(t, C(t), A_\beta(t), u(t)) = C(t) + \varepsilon(A_\beta(t), t)u^2(t)$, we introduce the Hamiltonian based on optimal control theory [9, 25],

$$H(t, \boldsymbol{x}(t), u(t), \boldsymbol{\Lambda}(t)) = f(t, C(t), A_\beta(t), u(t)) + \boldsymbol{\Lambda}(t)^T \boldsymbol{G}(t, \boldsymbol{x}(t), u(t)),$$

where the adjoint vector is $\boldsymbol{\Lambda} = (\Lambda_1, \Lambda_2, \Lambda_3, \Lambda_4, \Lambda_5)^T$. The state system is denoted by $\boldsymbol{x}' = \boldsymbol{G}(t, \boldsymbol{x}, u(t))$. Using Pontryagin's Maximum Principle [25], we obtain

$$\frac{d\Lambda_i}{dt} = -\frac{\partial H}{\partial x_i},$$

and the system of adjoint equations with its final time conditions,

$$\begin{cases} \dfrac{d\Lambda_1}{dt} = -\varepsilon_0 e^{-\gamma t} u^2 + \left( 2\dfrac{\lambda_{A_\beta}}{K_{A_\beta}} A_\beta - \lambda_{A_\beta} + u \right)\Lambda_1 - \lambda_\tau \left(1 - \dfrac{\tau_p}{K_\tau}\right)\Lambda_2 \\[2ex] \dfrac{d\Lambda_2}{dt} = \dfrac{\lambda_\tau A_\beta}{K_\tau}\Lambda_2 - \lambda_{N\tau_\rho}\left(1 - \dfrac{N}{K_N}\right)\Lambda_4 - \lambda_{C\tau}\left(1 - \dfrac{C}{K_C}\right)\Lambda_5 \\[2ex] \dfrac{d\Lambda_3}{dt} = -\lambda_{N\tau_o}\left(1 - \dfrac{N}{K_N}\right)\Lambda_4 \\[2ex] \dfrac{d\Lambda_4}{dt} = \dfrac{\lambda_{N\tau_o}\tau_0 + \lambda_{N\tau_\rho}\tau_\rho}{K_N}\Lambda_4 - \lambda_{CN}\left(1 - \dfrac{C}{K_C}\right)\Lambda_5 \\[2ex] \dfrac{d\Lambda_5}{dt} = -1 + \dfrac{\lambda_{CN}N}{K_C}\Lambda_5 \end{cases} \tag{12}$$

with $\Lambda_1(T_2) = \alpha_1$, $\Lambda_2(T_2) = \Lambda_3(T_2) = \Lambda_4(T_2) = 0$, and $\Lambda_5(T_2) = \alpha_2$.

On the interior of the control set, the optimal anti-amyloid-beta therapy $u^\star(t)$ satisfies the optimality equation

$$\frac{\partial H}{\partial u} = f_u + \boldsymbol{\Lambda}^T \boldsymbol{G}_u = 2\varepsilon u(t) - A_\beta(t)\Lambda_1(t) = 0 \Rightarrow u^\star(t) = \frac{\Lambda_1(t)A_\beta(t)}{2\varepsilon} \tag{13}$$

Then applying the bounds on the controls, we obtain the optimal control characterization,

$$u^\star(t) = \max\left[ u_{max}, \min\left[ 0, \frac{\Lambda_1(t)A_\beta(t)}{2\varepsilon} \right] \right]. \tag{14}$$

The optimality system consists of the state differential equations, (6) and (10) and the adjoint Eq (12), together with the optimal control characterization (14). Since the state equations have initial conditions and are coupled the adjoint equations with final time conditions, we use an iterative method, called the forward-backward sweep algorithms (shown in **Algorithm 2**) to solve the optimality system [9].

**Algorithm 2** Solving the optimality system

**Input** personalized parameter values and initial values for each
 patient.
1: Initialize the control $u(t)$, as a zero function;
2: Compute $\boldsymbol{x}$ by solving forward the state Eq (6) using the control
 $u(t)$;
3: Compute $\boldsymbol{\Lambda}(t)$ by solving backwards the adjoint Eq (12) using the
 states and the control;
4: Compute the new $u(t)$ by using the optimal control characterization
 (14) and update the control function as a convex combination of the
 previous control and the new control;

```
5: Compute the relative error of states, adjoints and the control.
   Continue to repeat steps 2-4, until the error is small.
```

## Results

### Personalized parameters

In order to estimate the parameters more accurately, we use the available patient data in ADNI-1 dataset with at least three longitudinal datapoints for each biomarker and take $T_0 =$ 50, given that the smallest age across the dataset is 54. The parameter estimation for selected patients in each group (AD, cognitively normal (CN), LMCI) are illustrated in Fig 3. The parameter values for each group are shown in Table 1, with relative error given by

$$\sqrt{\frac{1}{n}\sum_{i=1}^{n}\frac{(x(t_i) - \tilde{x}_i)^2}{\tilde{x}_i^2}}$$

where $x(t_i)$ is the model value of the biomarker while $\tilde{x}_i$ is the clinical measurement at age $t_i$.

**Parameter estimation of $u_{max}$.**   Based on the aducanumab data released by Biogen [26], there are two groups: low dosage and high dosage injections. The low dosage group for aducanumab was administered the drug 14 times, each treatment was 3 or 6 mg/kg. The cumulative dose at week 78 was 56 mg/kg to 98 mg/kg. The amyloid PET assessment was evaluated at week 78 and was decreased 16.5% comparing to the baseline. In this case, we consider

$$\frac{dA_\beta}{dt} = -u_{max}A_\beta \text{ which implies that } A_\beta(t) = A_\beta(0)e^{-u_{max}t}.$$

Accordingly, we have

$$u_{max} = -\frac{ln(0.835)}{78} = 2.31 \times 10^{-3}/\text{week}.$$

Similarly, the high dosage group was given 6-10 mg/kg aducanumab each time and received 116-153 mg/kg cumulatively at week 78. The amyloid PET assessment was decreased 27.2% at

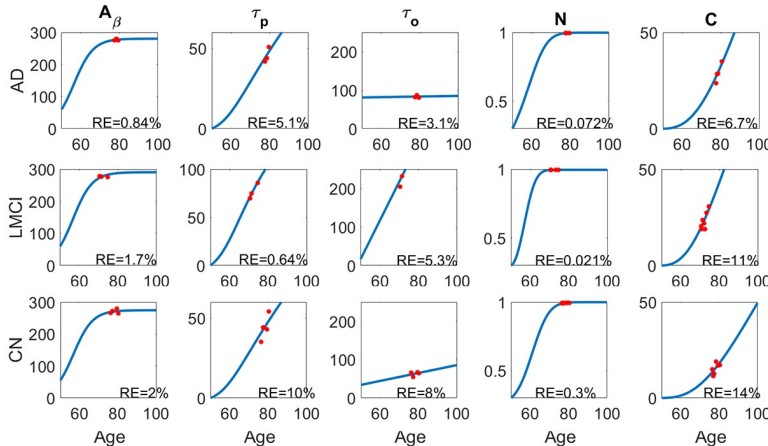

**Fig 3. The parameter fitting of the ODE model for one subject in each group.** The AD patient is female with age 84.7 (upper, subject # is AD4), the LMCI patient is male with age 82.8 (middle, subject # is MCI15), and the CN patient is female with age 81.8 (lower). The relative errors (RE) for each biomarker are also shown in each panel.

**Table 1. The mean initial conditions and parameter values for AD, LMCI and CN groups with relative errors for each of the biomarkers (n is the number of subjects).** The values are mean ± standard deviation (std).

| Descriptions | | AD group (n = 10) | LMCI group (n = 32) | CN group (n = 7) |
|---|---|---|---|---|
| Initial conditions | $A_0$ | 36.03 ± 26.52 | 41.57 ± 24.23 | 44.92 ± 24.54 |
| | $\tau_{p0}$ | 12.38 ± 14.47 | 4.21 ± 7.68 | 3.69 ± 6.15 |
| | $\tau_{o0}$ | 66.70 ± 58.57 | 28.66 ± 33.13 | 24.25 ± 26.98 |
| | $N_0$ | 0.26 ± 0.08 | 0.48 ± 0.26 | 0.42 ± 0.10 |
| | $C_0$ | 3.68 ± 8.30 | 6.03 ± 6.56 | 2.58 ± 2.60 |
| Parameter values | $\lambda_{A_\beta}$ | $(18.35 \pm 3.11) \times 10^{-2}$ | $(16.12 \pm 5.03) \times 10^{-2}$ | $(16.82 \pm 5.52) \times 10^{-2}$ |
| | $K_{A_\beta}$ | 259.44 ± 13.21 | 264.99 ± 74.69 | 276.21 ± 88.29 |
| | $\lambda_\tau$ | 0.15 ± 0.16 | 0.08 ± 0.12 | 0.12 ± 0.17 |
| | $K_\tau$ | 123.35 ± 81.63 | 131.66 ± 75.89 | 126.53 ± 91.31 |
| | $\lambda_{\tau_o}$ | 1.15 ± 1.70 | 1.74 ± 2.08 | 0.87 ± 0.66 |
| | $\lambda_{N\tau_o}$ | $(3.75 \pm 1.22) \times 10^{-4}$ | $(4.24 \pm 1.03) \times 10^{-4}$ | $(4.41 \pm 0.89) \times 10^{-4}$ |
| | $\lambda_{N\tau_p}$ | $(6.90 \pm 1.49) \times 10^{-3}$ | $(7.37 \pm 1.07) \times 10^{-3}$ | $(7.24 \pm 1.73) \times 10^{-3}$ |
| | $K_N$ | 1.00 ± 0.01 | 1.02 ± 0.05 | 1.03 ± 0.07 |
| | $\lambda_{CN}$ | 1.67 ± 2.40 | 1.26 ± 1.99 | 3.16 ± 3.06 |
| | $\lambda_{C\tau}$ | 3.83 ± 8.00 | 1.93 ± 3.91 | 2.48 ± 3.94 |
| | $K_C$ | 169.48 ± 63.35 | 129.40 ± 84.31 | 59.89 ± 80.03 |
| Relative errors | $A_\beta$ | 2.83 ± 1.21% | 11.31 ± 12.60% | 4.98 ± 3.56% |
| | $\tau_p$ | 8.44 ± 6.41% | 15.11 ± 11.02% | 12.25 ± 6.51% |
| | $\tau_o$ | 11.01 ± 6.24% | 19.28 ± 15.02% | 18.08 ± 7.94% |
| | $N$ | 0.17 ± 0.20% | 0.58 ± 0.63% | 0.79 ± 1.24% |
| | $C$ | 10.25 ± 4.37% | 16.00 ± 7.59% | 15.04 ± 3.79% |

week 78. Then

$$u_{max} = -\frac{ln(0.728)}{78} = 4.07 \times 10^{-3}/\text{week}.$$

We also estimate the clearance rate of donanemab by using the data in [27]. In particular, the amyloid plaque level, assessed by florbetapir PET relative specific uptake values (SUVr), is reduced 84.13 from 107.6 after a 76-week treatment. Similarly, we compute the maximum clearance rate as

$$u_{max} = -\frac{ln(23.47/107.6)}{76} = 2 \times 10^{-2}/\text{week}.$$

**Parameters in the Side-effect function $\varepsilon$.** We take $\alpha_1 = \alpha_2 = 1$ in the objective function (11). According to phase 3 studies of aducanumab [28], the dose regimen reaches the maximum dose after 10 and 25 weeks for the Low and High dose groups, respectively. Thus we estimate $\varepsilon_0$ and $\gamma$ by taking

$$u^*(10) \approx \frac{1}{2\varepsilon_0 e^{-10\gamma}} = 2.31 \times 10^{-3}/\text{week} \quad \text{and} \quad u^*(25) \approx \frac{1}{2\varepsilon_0 e^{-25\gamma}} = 4.07 \times 10^{-3}/\text{week}$$

which yields $\varepsilon_0 \approx 5$ and $\gamma \approx 2$.

**Numerical results.** We perform the personalized optimal control for each subject in both AD and LMCI groups with estimated parameters that are shown in Table 1. For each subject,

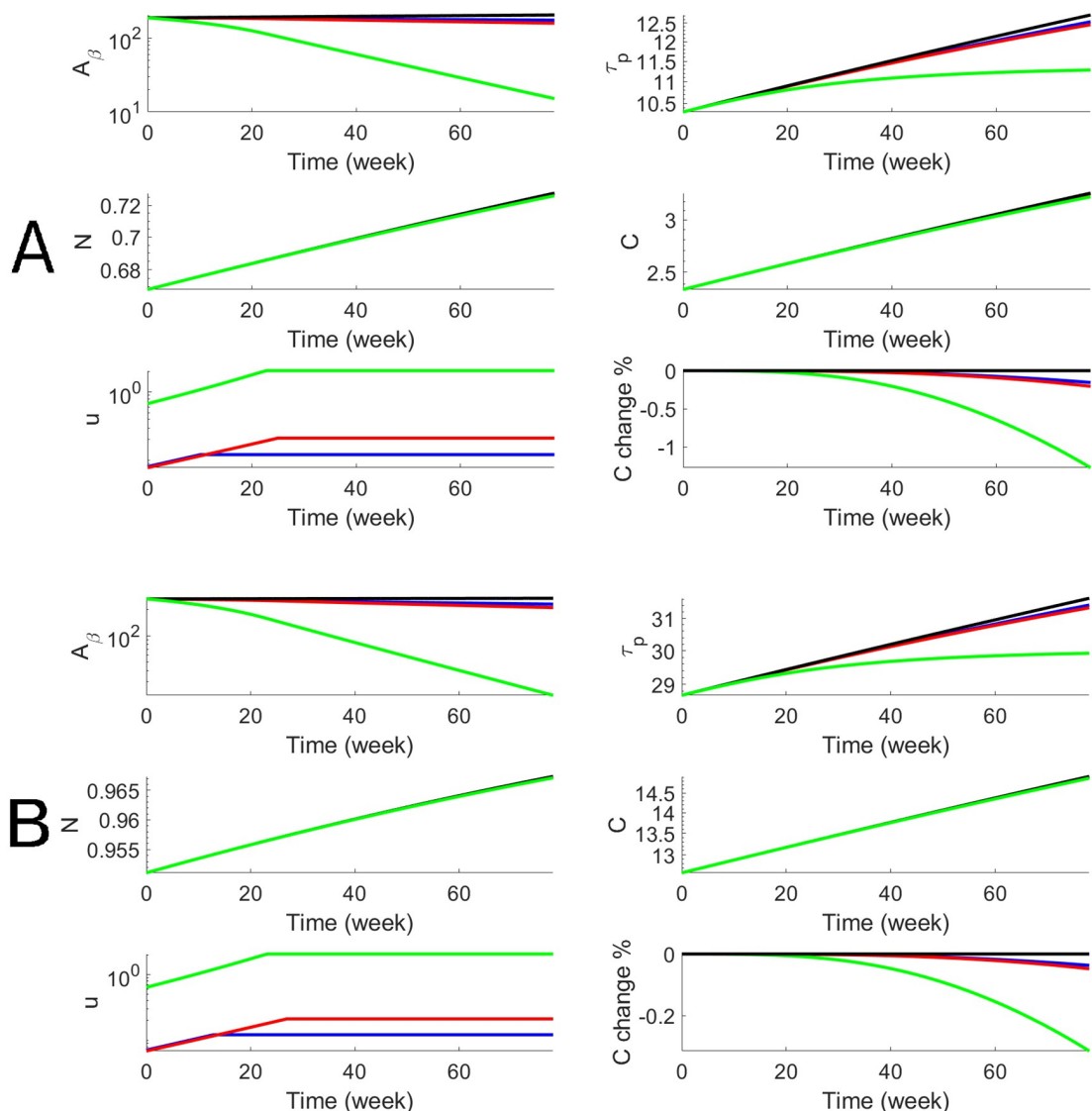

**Fig 4. Numerical results of the optimal anti-beta therapy for an AD patient (subjective # is AD4) for 78 weeks.** Blue and red curves stand for Aducanumab with low and high doses respectively while green curves stand for Donanemab. The treatment age starts at 60 (Panel A) and 70 (Panel B). The objective functional values defined in (8) are 187.5 (A, blue), 177.1 (A, red), 33.7 (A, green), 271.6 (B, blue), 258.6 (B, red), and 71.3 (B, green).

we have personalized optimal control for both a 78-week treatment and a 10-year treatment with low and high dosages. To illustrate the dynamics of biomarkers and the optimal drug dosage, we use one AD subject (Subject # is AD4) to show both short-term and long-term treatments in Figs 4 and 5. The efficacy of donanemab on both $A_\beta$ and p-tau is higher than aducanumab. The effect on cognitive decline, $C$, is modest for aducanumab while the effect of donanemab is more significant. In order to better assess the efficacy, we define the cognitive percentage change as $\dfrac{C(t) - C_0(t)}{C_0(t)} \times 100\%$ where $C(t)$ is cognitive decline with treatment and $C_0(t)$ is cognitive decline without treatment. Thus the maximum effects of donanemab and

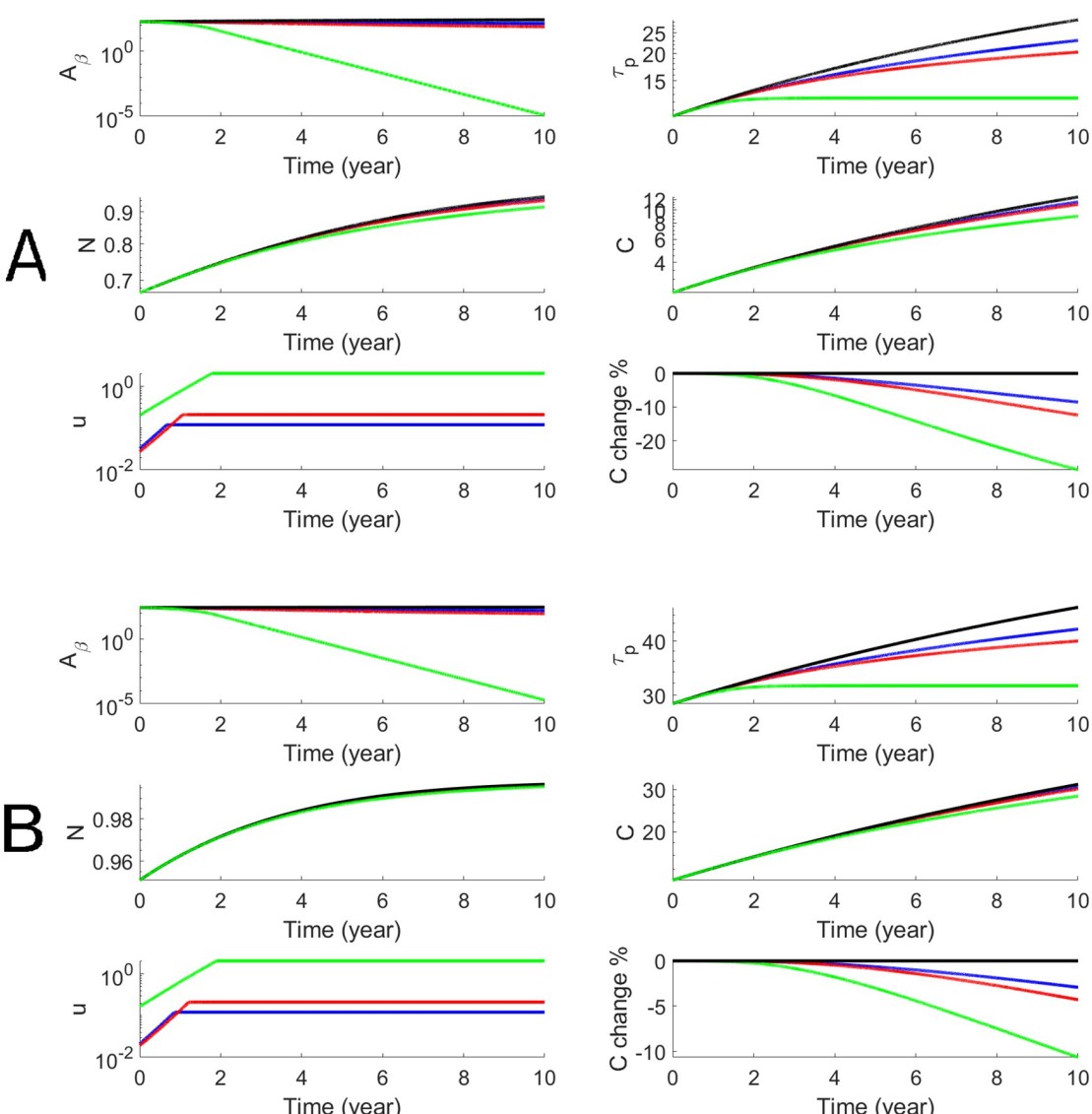

**Fig 5. Numerical results of the optimal anti-beta therapy for an AD patient (subjective # is AD4) for 10 years.** The treatment age starts at 60 (Panel A) and 70 (Panel B). The objective functional values defined in (8) are 216.2 (A, blue), 156.5 (A, red), 67.3 (A, green), 397.8 (B, blue) 335.1 (B, red), and 234.7 (B, green).

aducanumab are 8% and 20% less cognitive decline when the long-term treatment starts at Age 60. We define the cognitive percentage change at the end of the treatment as $\dfrac{C(T_2) - C_0(T_2)}{C_0(T_2)} \times 100\%$ and summarize the cognitive percentage change for the AD group (n = 10) in Table 2. It shows that the maximum effects of aducanumab and donanemab have median values of 5.2% and 13.1%, for the AD group with the long-term treatment. Similarly, we illustrate the personalized optimal treatment for an LMCI subject (subject # is MCI15) for the short-term treatment in Fig 6 and for the long-term treatment in Fig 7. The effect of the personalized optimal donanemab and aducanumab treatments on cognitive decline for the

**Table 2. The percentage change of the cognitive decline by the end of the treatment period for the AD group with both the 78-week (upper) and 10-year (lower) treatments.** The treatment age starts at 60 and 70 with both low and high dosages. "NR" stands for "No response" which is defined by percentage change less than $10^{-7}$.

| | Subject # | Starting at Age 60 | | | Starting at Age 70 | | |
|---|---|---|---|---|---|---|---|
| | | Aducanumab | | Donanemab | Aducanumab | | Donanemab |
| | | Low dose | High dose | | Low dose | High dose | |
| 78 week | AD1 | 0.13 | 0.17 | 1 | $2.5 \times 10^{-2}$ | $3.3 \times 10^{-2}$ | 0.22 |
| | AD2 | 0.15 | 0.19 | 1.2 | $3.6 \times 10^{-2}$ | $4.6 \times 10^{-2}$ | 0.30 |
| | AD3 | $1.2 \times 10^{-3}$ | $2.4 \times 10^{-3}$ | $1.2 \times 10^{-2}$ | $5.3 \times 10^{-4}$ | $7.8 \times 10^{-4}$ | $1.2 \times 10^{-2}$ |
| | AD4 | $2.6 \times 10^{-3}$ | $3.7 \times 10^{-3}$ | $2 \times 10^{-2}$ | $1.1 \times 10^{-3}$ | $1.6 \times 10^{-3}$ | $9.4 \times 10^{-3}$ |
| | AD5 | 0.15 | 0.2 | 1.3 | $3.8 \times 10^{-2}$ | $4.9 \times 10^{-2}$ | 0.32 |
| | AD6 | $1.2 \times 10^{-6}$ | $2.4 \times 10^{-6}$ | $5 \times 10^{-5}$ | $2.3 \times 10^{-7}$ | $5.3 \times 10^{-7}$ | $1.5 \times 10^{-5}$ |
| | AD7 | $1 \times 10^{-5}$ | $1.3 \times 10^{-5}$ | $9.9 \times 10^{-4}$ | NR | NR | NR |
| | AD8 | $3.5 \times 10^{-5}$ | $4.6 \times 10^{-5}$ | $3.4 \times 10^{-4}$ | NR | NR | NR |
| | AD9 | 0.16 | 0.21 | 1.3 | $3.7 \times 10^{-2}$ | $4.8 \times 10^{-2}$ | 0.32 |
| | AD10 | 0.13 | 0.18 | 1 | $4 \times 10^{-2}$ | $5.2 \times 10^{-2}$ | 0.31 |
| | Median | $6.6 \times 10^{-2}$ | $8.6 \times 10^{-2}$ | 0.51 | $1.3 \times 10^{-2}$ | $1.7 \times 10^{-2}$ | 0.11 |
| 10 year | AD1 | 6.6 | 9.9 | 25 | 1.8 | 2.6 | 7.2 |
| | AD2 | 8.1 | 12 | 27 | 2.9 | 4.2 | 10 |
| | AD3 | $5.5 \times 10^{-7}$ | $5.7 \times 10^{-7}$ | $7.9 \times 10^{-7}$ | NR | $1.6 \times 10^{-7}$ | $1.9 \times 10^{-7}$ |
| | AD4 | 0.36 | 0.54 | 1.1 | 0.028 | 0.048 | 0.13 |
| | AD5 | 8.4 | 12 | 28 | 3 | 4.4 | 11 |
| | AD6 | $1.2 \times 10^{-5}$ | $2.2 \times 10^{-5}$ | $5.3 \times 10^{-5}$ | NR | NR | $1.8 \times 10^{-7}$ |
| | AD7 | $1.5 \times 10^{-4}$ | $1.9 \times 10^{-4}$ | $2.3 \times 10^{-4}$ | NR | NR | NR |
| | AD8 | $1.9 \times 10^{-4}$ | $2.8 \times 10^{-4}$ | $1.4 \times 10^{-3}$ | NR | NR | $1.2 \times 10^{-7}$ |
| | AD9 | 8.5 | 12 | 29 | 2.9 | 4.3 | 11 |
| | AD10 | 9.2 | 13 | 26 | 3.8 | 5.3 | 11 |
| | Median | 3.5 | 5.2 | 13.1 | 0.9 | 1.3 | 3.6 |

LMCI group is summarized in Table 3 for the short-term treatment, and Table 4 for the long-term treatment. The maximum effects of aducanumab and donanemab have median values of 5.3% and 13%, respectively, for the LMCI group with the long-term treatment.

## Discussion

In this paper we develop a data-driven modeling approach to model the progression of AD biomarkers which integrates AD pathophysiology and clinical data. We develop and refine a mathematical model in terms of a system of ODEs to describe progression of the AD bio-marker cascade. By using available biomarker data in a large multi-center natural history trial, ADNI, we parametrize the ODE model to build a personalized model for each patient. In order to solve the non-convex optimization arising from parameter estimation, we develop an "equation-by-equation" approach to calibrate the cascade model. The average relative errors of the fitting process are ~10% for AD group and ~15% for CN and LMCI groups.

We also perform an in-silico personalized optimal treatment study by adding a control function to model anti-amyloid-beta treatment. By maximizing treatment effects on cognitive decline and minimizing the side effects of anti-amyloid-beta therapy, we develop the first computational framework to simulate an optimal treatment regimen via optimal control theory. We represent the side effects of anti-amyloid-beta therapy as a function of both the amyloid beta concentration, dose and treatment duration. The results show that the optimal

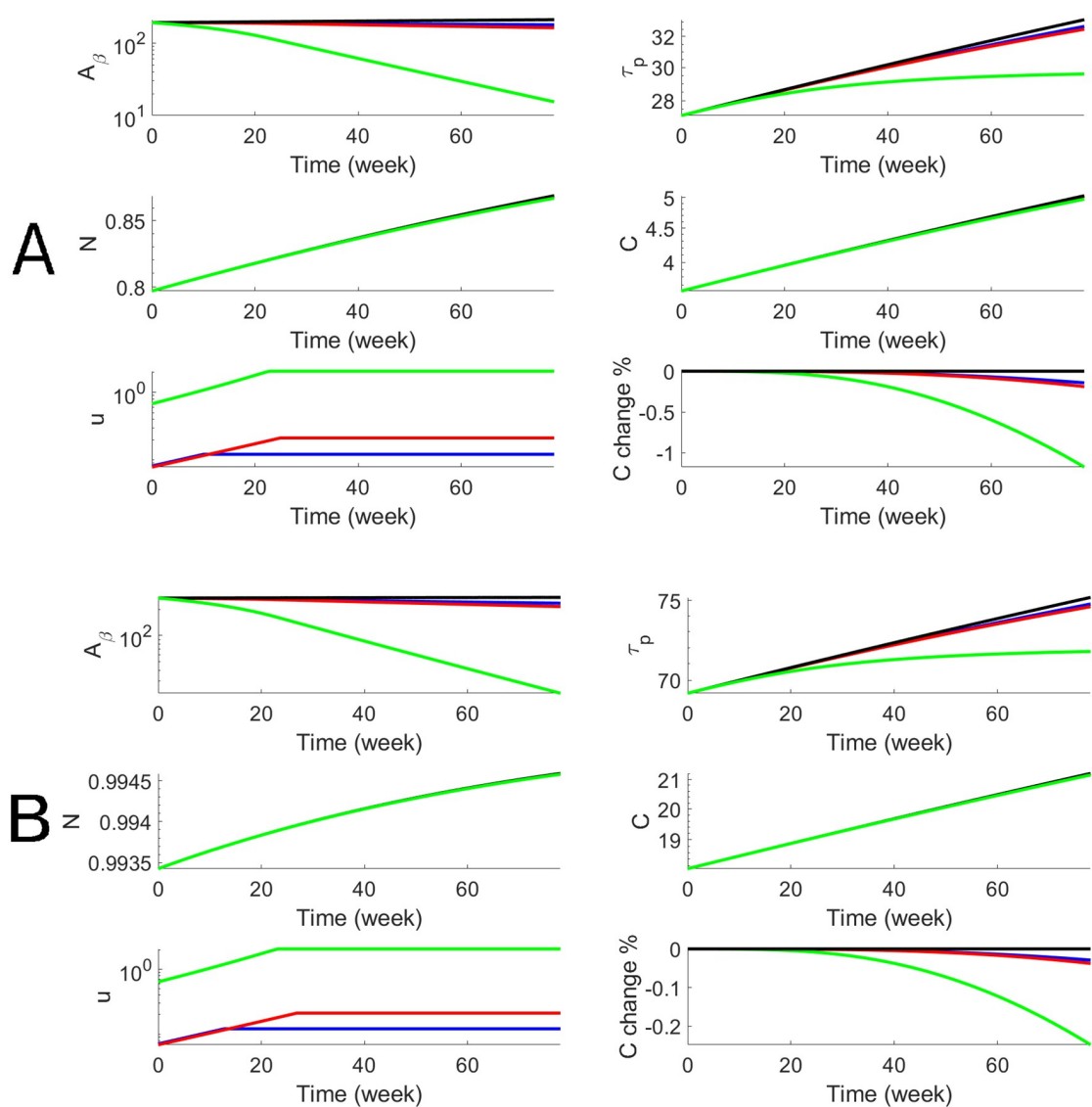

**Fig 6. Numerical results of the optimal anti-beta therapy for the LMCI patient (subject # is MCI15) for 78 weeks.** The objective functional values defined in (8) are 195.5 (A, blue), 184.8 (A, red), 38.2 (A, green), 294.7 (B, blue), 281.3 (B, red), and 87.6 (B, green).

treatment regimen gradually increases dose until it reaches as maximum dosage steady state. It approximates the dosage scheduling in the aducanumab clinical trail conducted by Biogen [22]. In agreement with the data provided by Biogen for the 78-week clinical trial, amyloid beta concentration is decreased by 27% for high dosage and 16% for low dosage. A decrease of p-tau concentration is observable for the 10-year optimal treatment study. In keeping with actual clinical trial results of these agents administered in MCI and AD subjects, anti-amyloid-beta treatment has a modest mitigation effect on cognitive decline for both short-term and long-term treatments. Our study shows that aducanumab's efficacy as a treatment for cognitive dysfunction in AD is limited even by an optimal dosage regimen with a long-term treatment. However, donanemab's efficacy is higher, according to the model, than that of aducanumab.

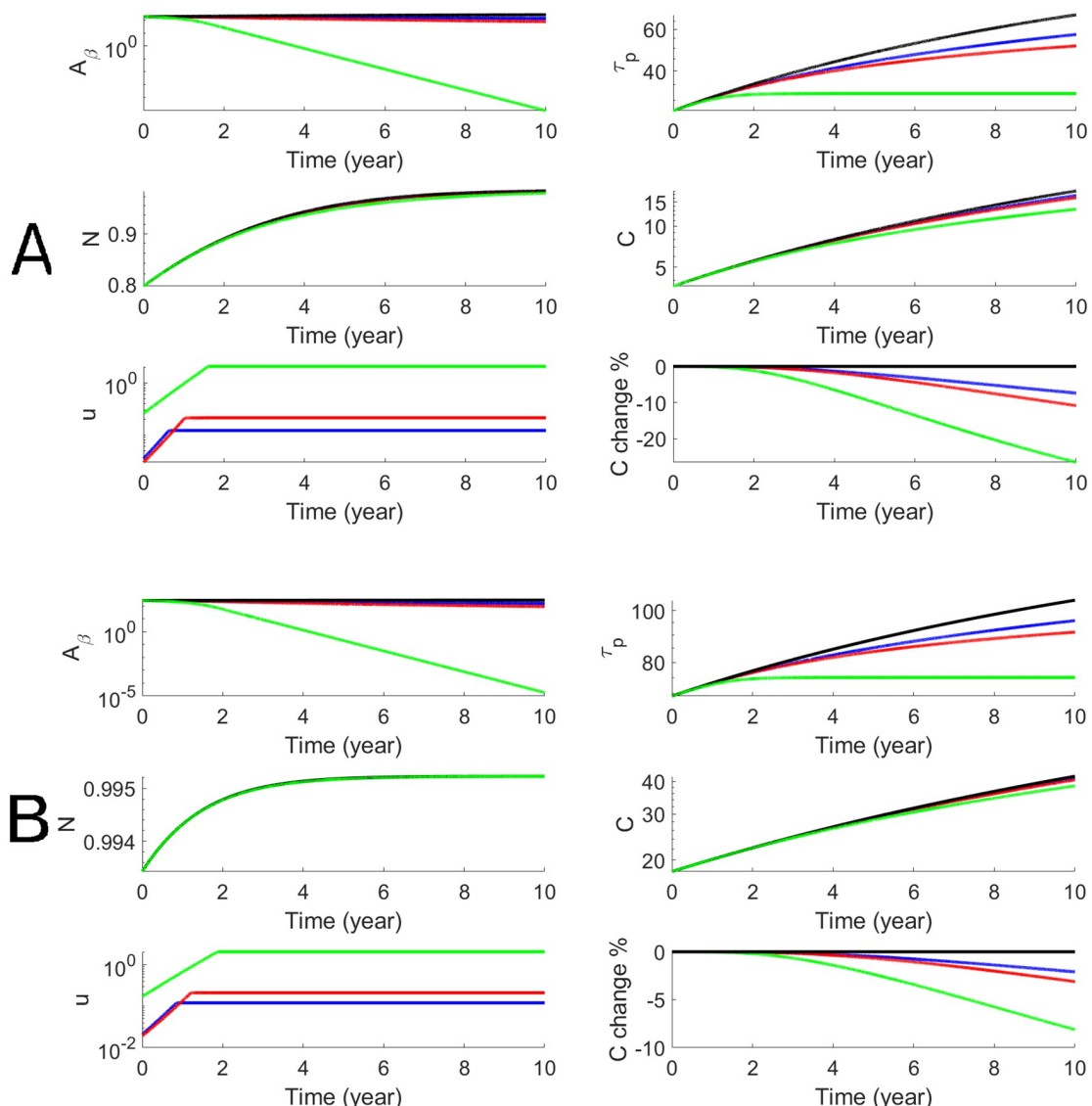

**Fig 7. Numerical results of the optimal anti-beta therapy for the LMCI patient (subject # is MCI15) for 10 years.** The objective functional values defined in (8) are 256.8 (A, blue), 194.7 (A, red), 100.2 (A, green), 493.8 (B, blue), 428.8 (B, red), and 324.8 (B, green).

The buildup of amyloid plaques in the brains of AD patients is thought to result from an imbalance between amyloid clearance and production [29]. Removing amyloid plaques via pharmoco-therapy accelerates amyloid clearance, but only for the duration of treatment, given that the factors leading to the native imbalance are not removed. For this reason, it is assumed that anti-amyloid-beta treatment, in the form of the current immunotherapies, will be necessary over remainder of a patient's lifetime for sustained disease management, similar to insulin therapy in a diabetic patient. We therefore simulated sustained therapy over the course of a decade, in addition to the typical clinical trial duration of 78 weeks.

In summary, we developed a novel modeling approach to provide a personalized optimal AD treatment plan for individual patients, using optimal control theory. This approach allows

**Table 3. The percentage change of the cognitive decline by the end of the 78-week treatment period for the LMCI group.**

| Subject # | Starting at Age 60 | | | Starting at Age 70 | | |
|---|---|---|---|---|---|---|
| | Aducanumab | | Donanemab | Aducanumab | | Donanemab |
| | Low dose | High dose | | Low dose | High dose | |
| MCI1 | 0.1 | 0.14 | 0.86 | $9.7 \times 10^{-3}$ | $1.3 \times 10^{-2}$ | $8.4 \times 10^{-2}$ |
| MCI2 | $5.9 \times 10^{-2}$ | $7.9 \times 10^{-2}$ | 0.45 | $2.5 \times 10^{-2}$ | $3.3 \times 10^{-2}$ | 0.2 |
| MCI3 | 0.12 | 0.16 | 1 | $1.6 \times 10^{-2}$ | $2.1 \times 10^{-2}$ | 0.14 |
| MCI4 | $4.3 \times 10^{-6}$ | $6.1 \times 10^{-6}$ | $3.3 \times 10^{-5}$ | NR | NR | NR |
| MCI5 | $1.6 \times 10^{-3}$ | $2.2 \times 10^{-3}$ | $1.3 \times 10^{-2}$ | $4.7 \times 10^{-6}$ | $6.3 \times 10^{-6}$ | $4.7 \times 10^{-5}$ |
| MCI6 | $1.9 \times 10^{-3}$ | $2.4 \times 10^{-3}$ | $1.8 \times 10^{-2}$ | $7.6 \times 10^{-7}$ | $9.7 \times 10^{-7}$ | $7.8 \times 10^{-6}$ |
| MCI7 | $1.1 \times 10^{-2}$ | $1.4 \times 10^{-2}$ | $8.9 \times 10^{-2}$ | $9.6 \times 10^{-3}$ | $1.3 \times 10^{-2}$ | $8.2 \times 10^{-2}$ |
| MCI8 | 0.14 | 0.19 | 1.2 | $2.9 \times 10^{-2}$ | $3.8 \times 10^{-2}$ | 0.25 |
| MCI9 | $4.1 \times 10^{-4}$ | $5.8 \times 10^{-4}$ | $3.2 \times 10^{-3}$ | $2.4 \times 10^{-4}$ | $3.3 \times 10^{-4}$ | $2 \times 10^{-3}$ |
| MCI10 | $8.3 \times 10^{-7}$ | $1.1 \times 10^{-6}$ | $6.8 \times 10^{-6}$ | NR | NR | NR |
| MCI11 | $2.5 \times 10^{-4}$ | $3.5 \times 10^{-4}$ | $2 \times 10^{-3}$ | $10^{-15}$ | $2 \times 10^{-5}$ | NR |
| MCI12 | $1.2 \times 10^{-2}$ | $1.5 \times 10^{-2}$ | $9.9 \times 10^{-2}$ | $2.8 \times 10^{-4}$ | $3.6 \times 10^{-4}$ | $2.5 \times 10^{-3}$ |
| MCI13 | 0.11 | 0.15 | 0.92 | $1.2 \times 10^{-2}$ | $1.6 \times 10^{-2}$ | 0.1 |
| MCI14 | $4.8 \times 10^{-3}$ | $6.5 \times 10^{-3}$ | $3.7 \times 10^{-2}$ | $5.7 \times 10^{-3}$ | $7.5 \times 10^{-3}$ | $4.5 \times 10^{-2}$ |
| MCI15 | NR | NR | NR | NR | NR | NR |
| MCI16 | $3.6 \times 10^{-2}$ | $5 \times 10^{-2}$ | 0.3 | $1.4 \times 10^{-4}$ | $1.9 \times 10^{-4}$ | $1.4 \times 10^{-3}$ |
| MCI17 | $7.4 \times 10^{-7}$ | $9.5 \times 10^{-7}$ | $8.5 \times 10^{-6}$ | NR | NR | NR |
| MCI18 | $1.7 \times 10^{-3}$ | $2.2 \times 10^{-3}$ | $1.4 \times 10^{-2}$ | $1.6 \times 10^{-3}$ | $2 \times 10^{-3}$ | $1.3 \times 10^{-2}$ |
| MCI19 | $6.2 \times 10^{-2}$ | $8.4 \times 10^{-2}$ | 0.47 | $3.7 \times 10^{-2}$ | $5.1 \times 10^{-2}$ | 0.29 |
| MCI20 | $5.1 \times 10^{-2}$ | $6.7 \times 10^{-2}$ | 0.42 | $1.4 \times 10^{-2}$ | $1.8 \times 10^{-2}$ | 0.12 |
| MCI21 | $2.3 \times 10^{-2}$ | $3.2 \times 10^{-2}$ | 0.18 | $2.2 \times 10^{-2}$ | $3 \times 10^{-2}$ | 0.17 |
| MCI22 | $1.9 \times 10^{-6}$ | $2.5 \times 10^{-6}$ | $2.1 \times 10^{-5}$ | NR | NR | NR |
| MCI23 | $4.5 \times 10^{-3}$ | $5.9 \times 10^{-3}$ | $3.7 \times 10^{-2}$ | $4.2 \times 10^{-3}$ | $5.4 \times 10^{-3}$ | $3.6 \times 10^{-2}$ |
| MCI24 | $1.2 \times 10^{-5}$ | $1.6 \times 10^{-5}$ | NR | NR | NR | NR |
| MCI25 | $4.4 \times 10^{-3}$ | $6.1 \times 10^{-3}$ | $3.7 \times 10^{-2}$ | $1.8 \times 10^{-5}$ | $2.4 \times 10^{-5}$ | $1.8 \times 10^{-4}$ |
| MCI26 | 0.13 | 0.17 | 1 | $2.4 \times 10^{-2}$ | $3.1 \times 10^{-2}$ | 0.2 |
| MCI27 | $7.6 \times 10^{-3}$ | $10^{-2}$ | $6.4 \times 10^{-2}$ | $6.3 \times 10^{-3}$ | $8.1 \times 10^{-3}$ | $5.3 \times 10^{-2}$ |
| MCI28 | $1.7 \times 10^{-3}$ | $2.3 \times 10^{-3}$ | $1.4 \times 10^{-2}$ | NR | NR | $3.1 \times 10^{-7}$ |
| MCI29 | 0.28 | 0.4 | 2.2 | 0.16 | 0.23 | 1.3 |
| MCI30 | 0.12 | 0.16 | 0.92 | $2.8 \times 10^{-2}$ | $3.8 \times 10^{-2}$ | 0.22 |
| MCI31 | 0.14 | 0.19 | 1.1 | $4 \times 10^{-2}$ | $5.2 \times 10^{-2}$ | 0.32 |
| MCI32 | 0.1 | 0.15 | 1 | $5 \times 10^{-2}$ | $4 \times 10^{-2}$ | 0.25 |
| Median | 0.12 | 0.16 | 0.92 | 0.16 | 0.23 | 0.21 |

us to integrate personal longitudinal biomarker data into the model by fitting the personalized parameters. This modeling approach, though a simplification, is based on current theories of AD pathophysiology which continue to undergo refinement. The optimal treatment takes into account the side effects of anti-amyloid-beta therapy, including amyloid-related imaging abnormalities (ARIA). Given the established framework, this approach can be easily extended to include other treatments, such as anti-tau therapy, as well as combined therapies, as more clinical trial data becomes available. Future directions include extending the current model to the spatiotemporal domain, by including spatial information from available imaging biomarkers, to evaluate the effects of treatment on whole-brain neuropathology and neurodegeneration. We will further validate and test the optimal treatment approach using other publically

**Table 4. The percentage change of the cognitive decline by the end of the 10-year treatment period for the LMCI group.**

| Subject # | Starting at Age 60 | | | Starting at Age 70 | | |
|---|---|---|---|---|---|---|
| | Aducanumab | | Donanemab | Aducanumab | | Donanemab |
| | Low dose | High dose | | Low dose | High dose | |
| MCI1 | 3.4 | 5.1 | 15 | 0.33 | 0.49 | 1.2 |
| MCI2 | 5.7 | 7.8 | 16 | 2.8 | 3.8 | 8.2 |
| MCI3 | 4.6 | 6.9 | 18 | 0.55 | 0.83 | 2 |
| MCI4 | NR | NR | NR | NR | NR | NR |
| MCI5 | 0.12 | 0.19 | 0.53 | $2.5 \times 10^{-5}$ | $3.3 \times 10^{-5}$ | $4.3 \times 10^{-5}$ |
| MCI6 | $1.4 \times 10^{-2}$ | $2.2 \times 10^{-2}$ | $3.4 \times 10^{-2}$ | $3.3 \times 10^{-6}$ | $5.3 \times 10^{-6}$ | $3.2 \times 10^{-6}$ |
| MCI7 | 1.9 | 2.7 | 5.7 | 1.3 | 1.9 | 4.2 |
| MCI8 | 7.4 | 11 | 27 | 2.1 | 3.1 | 8.1 |
| MCI9 | 0.12 | 0.17 | 0.32 | $1 \times 10^{-2}$ | $1.8 \times 10^{-2}$ | $2.9 \times 10^{-2}$ |
| MCI10 | NR | NR | $5.1 \times 10^{-7}$ | NR | NR | NR |
| MCI11 | NR | NR | $1.2 \times 10^{-6}$ | NR | NR | $2.2 \times 10^{-7}$ |
| MCI12 | $1.1 \times 10^{-7}$ | $5 \times 10^{-7}$ | $5.7 \times 10^{-6}$ | NR | NR | $3.5 \times 10^{-7}$ |
| MCI13 | 0.18 | 0.28 | 0.71 | $2.5 \times 10^{-3}$ | $4 \times 10^{-3}$ | $4.3 \times 10^{-2}$ |
| MCI14 | 3.6 | 5.4 | 15 | 0.26 | 0.4 | 0.88 |
| MCI15 | 1.1 | 1.5 | 2.7 | 1.1 | 1.5 | 2.7 |
| MCI16 | $1.8 \times 10^{-7}$ | $2.9 \times 10^{-7}$ | $1.9 \times 10^{-6}$ | NR | NR | NR |
| MCI17 | 0.89 | 1.6 | 6.5 | $4.2 \times 10^{-4}$ | $5.9 \times 10^{-4}$ | $1.5 \times 10^{-3}$ |
| MCI18 | $7.5 \times 10^{-7}$ | $9.2 \times 10^{-7}$ | $1.1 \times 10^{-6}$ | NR | NR | NR |
| MCI19 | 0.29 | 0.42 | 0.73 | 0.24 | 0.35 | 0.63 |
| MCI20 | 7.3 | 9.8 | 19 | 4.9 | 6.5 | 13 |
| MCI21 | 4 | 5.9 | 16 | 1 | 1.5 | 4.1 |
| MCI22 | 4.3 | 5.8 | 11 | 3.3 | 4.5 | 9.1 |
| MCI23 | $3.8 \times 10^{-6}$ | $4.7 \times 10^{-6}$ | $7.2 \times 10^{-5}$ | NR | NR | NR |
| MCI24 | 0.82 | 1.2 | 2.4 | 0.62 | 0.91 | 1.9 |
| MCI25 | NR | NR | NR | NR | NR | NR |
| MCI26 | 0.12 | 0.2 | 0.53 | $6.8 \times 10^{-5}$ | $1 \times 10^{-4}$ | $3.2 \times 10^{-3}$ |
| MCI27 | 6.3 | 9.3 | 24 | 1.7 | 2.5 | 6.6 |
| MCI28 | 1.2 | 1.8 | 4 | 0.88 | 1.3 | 2.9 |
| MCI29 | $9.1 \times 10^{-5}$ | $1.5 \times 10^{-4}$ | $1.4 \times 10^{-3}$ | NR | NR | NR |
| MCI30 | 21 | 31 | 55 | 8.8 | 14 | 31 |
| MCI31 | 7.9 | 11 | 23 | 2.2 | 3.1 | 6.8 |
| MCI32 | 9.2 | 13 | 27 | 3.5 | 5 | 11 |
| Median | 3.5 | 5.3 | 13 | 1.3 | 1.9 | 4.2 |

available datasets to verify the efficacy of anti-Abeta therapy. Moreover, when the data from the Aducanumab phase 3 studies become available, we will further calibrate and refine the in-silico anti-Abeta therapy model and test its efficacy.

## Author Contributions

**Conceptualization:** Suzanne Lenhart, Jeffrey R. Petrella.

**Data curation:** Jeffrey R. Petrella.

**Methodology:** Wenrui Hao, Suzanne Lenhart.

**Project administration:** Wenrui Hao.

**Resources:** Jeffrey R. Petrella.

**Supervision:** Jeffrey R. Petrella.

**Validation:** Suzanne Lenhart.

**Visualization:** Jeffrey R. Petrella.

**Writing – original draft:** Wenrui Hao.

**Writing – review & editing:** Suzanne Lenhart, Jeffrey R. Petrella.

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
