## [Decision Letter · Decision Letter 0]

20 May 2022

Dear Dr Hao,

Thank you very much for submitting your manuscript "Optimal Anti-amyloid-beta Therapy for Alzheimer’s Disease via a Personalized Mathematical Model" for consideration at PLOS Computational Biology. As with all papers reviewed by the journal, your manuscript was reviewed by members of the editorial board and by several independent reviewers. The reviewers appreciated the attention to an important topic. Based on the reviews, we are likely to accept this manuscript for publication, providing that you modify the manuscript according to the review recommendations.

Please carefully respond to all reviewer comments and address concerns within the manuscript.

Sincerely,

Adrianne Jenner

Associate Editor

PLOS Computational Biology

Virginia Pitzer

Deputy Editor-in-Chief

PLOS Computational Biology

[LINK]

Please carefully respond to all reviewer comments and address concerns within the manuscript.

Reviewer's Responses to Questions

**Comments to the Authors:**

Reviewer #1: In this paper, an ODE system consists of five equations governing the dynamics of Amyloid beta, Phosphorylated tau, nonamyloid-dependent tau, neurodegeneration, and cognitive decline is used to study Alzheimer’s disease (AD) treatment. In particular, it incorporates four AD clinical markers to determine individual parameters for the ODE system sequentially. The AD interventions are studied by adding a degradation term in the equation of Amyloid beta. The objective is to find a treatment to minimize both amyloid and cognitive impairment at the end of the treatment while keeping the cognitive impairment and side-effects over the treatment interval minimal.

The paper is well written and is easy for readers to understand and follow. The overall goals are well articulated and authors proposed methods based on ODE model, parameter fitting, and numerical optimization to find the optimal therapy. Thus I would recommend the publication after a minor revision. Here are some questions that authors may address.

Q1: Even though AD clinical markers are used to determine individual parameters, some dataset only has three longitudinal datapoints in a short range of time. The starting time is chosen as T_0 = 50. Will the fitting results vary a lot with respect to the choice of the initial time? In the paragraph, it was mentioned that some longitudinal dataset is available for up to 10 years. Does the parameter fitting work well for these kinds of dataset?

Q2: For Table 2, how can one tell whether they are for 78-week or 10-year treatments? Top versus bottom? As the numerical values ranges roughly from 10^-7 to 29, the average may not be a good indicator. Maybe use median instead. Most readers will be benefited from understanding why the decline varies widely. In what situation, the decline will be small? In Table 3 and Table 4, some values are as small as 10^-16 which is about the machine precision. This raises the concern of the accuracy of the numerical methods. Can authors address that? Are the results reliable?

Here are some minor suggestions:

Keywords: “Alzheimer's” instead of “alzheimer's”

On page 9, one of the “%” in relative errors for A_{\\beta} is misallocated.

On page 10, caption in Fig.3: “with age 84.7” instead of “with 84.7”.

On page 11, line 172: “are in average 5.9%” instead of “are 5.9%”. Is this for 78-week treatment? Also check the last sentence in this paragraph. Be specific in discussing the results that you have.

Reviewer #2: The review is uploaded as an attachment. Please address all my (minor) concerns.

Reviewer #3: The authors formulate a mathematical model which enables them to simulate personalized clinical trials of the anti-amyloid medicines Aducanumab and Donanemab. The data of the selected patients are taken from the ADNI data bank. The conclusions of the simulations seem to agree with actual clinical trials, in the sense that the amount of amyloid is reduced substantially but the effects on cognitive decline are marginal.

The basic mathematical equations are surprisingly simple (for example, spatial effects are totally neglected). Therefore the model contains very few parameters which, due to the cascade-structure of the equations, can be easily estimated from known biomarkers. The personalized character of the in silico trials is modeled through relatively simple optimization techniques. It is at least curious that such a simple mathematical set-up leads to results which confirm actual clinical trials. Without any doubt this makes the paper interesting.

To fully judge the importance and level of reliability of the model, it would be important to have information about the optimality of single personalized trials. Do the authors have any indications for the reliability of the optimization of the trial for single patients? If not, do they have any argument to convince the reader of such reliability, or, at least, can they indicate how to handle this issue in the future? This would make the paper more valuable.

**Have the authors made all data and (if applicable) computational code underlying the findings in their manuscript fully available?**

Reviewer #1: **No: **The authors plan to make codes available upon the acceptance of the article

Reviewer #2: Yes

Reviewer #3: Yes

PLOS authors have the option to publish the peer review history of their article (what does this mean?). If published, this will include your full peer review and any attached files.

Reviewer #1: No

Reviewer #2: No

Reviewer #3: No

Figure Files:

Data Requirements:

Reproducibility:

References:

---

## [Decision Letter · Decision Letter 1]

28 Jul 2022

Dear Dr Hao,

Thank you very much for submitting your manuscript "Optimal Anti-amyloid-beta Therapy for Alzheimer’s Disease via a Personalized Mathematical Model" for consideration at PLOS Computational Biology. As with all papers reviewed by the journal, your manuscript was reviewed by members of the editorial board and by several independent reviewers. The reviewers appreciated the attention to an important topic. Based on the reviews, we are likely to accept this manuscript for publication, providing that you modify the manuscript according to the review recommendations.

Please make the minor changes that the reviewer suggests (below) before we accept can accept manuscript for publication. Also, please ensure that the code is published and available at this stage, as dictated by our code-sharing policy.

Sincerely,

Adrianne Jenner

Associate Editor

PLOS Computational Biology

Virginia Pitzer

Deputy Editor-in-Chief

PLOS Computational Biology

[LINK]

Can the authors please make the minor changes that the reviewer accepted before we accept their manuscript for publication.

Reviewer's Responses to Questions

**Comments to the Authors:**

Reviewer #1: The authors have addressed reviewers’ suggestions. The publication is recommended after the minor revision. Here are some more minor suggestions.

On page 2, line 39: remove the space after taupathology.

On page 8, line 148: add a space after (6).

It seems that \\lambda_{\\tau A_\\beta} and \\lambda_{\\tau} refer to the same parameter. It is better to make it consistent across the manuscript.

Reviewer #2: The authors have satisfactory responded to all reviewer comments and concerns.

**Have the authors made all data and (if applicable) computational code underlying the findings in their manuscript fully available?**

Reviewer #1: **No: **The code will be published upon the acceptance of the article.

Reviewer #2: Yes

PLOS authors have the option to publish the peer review history of their article (what does this mean?). If published, this will include your full peer review and any attached files.

Reviewer #1: No

Reviewer #2: No

Figure Files:

Data Requirements:

Reproducibility:

References:

---

## [Editor Report · Decision Letter 2]

10 Aug 2022

Dear Dr Hao,

We are pleased to inform you that your manuscript 'Optimal Anti-amyloid-beta Therapy for Alzheimer’s Disease via a Personalized Mathematical Model' has been provisionally accepted for publication in PLOS Computational Biology.

Best regards,

Adrianne Jenner

Associate Editor

PLOS Computational Biology

Virginia Pitzer

Deputy Editor-in-Chief

PLOS Computational Biology

---

## [Editor Report · Acceptance letter]

25 Aug 2022

PCOMPBIOL-D-22-00432R2 

Optimal Anti-amyloid-beta Therapy for Alzheimer’s Disease via a Personalized Mathematical Model

Dear Dr Hao,

I am pleased to inform you that your manuscript has been formally accepted for publication in PLOS Computational Biology. Your manuscript is now with our production department and you will be notified of the publication date in due course.

With kind regards,

Anita Estes
